# Australia’s Disability Employment Services Program: Participant Perspectives on Factors Influencing Access to Work

**DOI:** 10.3390/ijerph182111485

**Published:** 2021-10-31

**Authors:** Alexandra Devine, Marissa Shields, Stefanie Dimov, Helen Dickinson, Cathy Vaughan, Rebecca Bentley, Anthony D. LaMontagne, Anne Kavanagh

**Affiliations:** 1Melbourne School of Population and Global Health, University of Melbourne, Melbourne 3010, Australia; marissa.shields@unimelb.edu.au (M.S.); stefanie.dimov@unimelb.edu.au (S.D.); cmvaug@unimelb.edu.au (C.V.); brj@unimelb.edu.au (R.B.); a.kavanagh@unimelb.edu.au (A.K.); 2School of Business, University of New South Wales, Canberra 2610, Australia; h.dickinson@adfa.edu.au; 3School of Health and Social Development, Deakin University, Melbourne 3125, Australia; tony.lamontagne@deakin.edu.au

**Keywords:** disability employment programs, vocational, non-vocational and structural barriers to work, paid employment for people with disability

## Abstract

Disability employment programs play a key role in supporting people with disability to overcome barriers to finding and maintaining work. Despite significant investment, ongoing reforms to Australia’s Disability Employment Services (DES) are yet to lead to improved outcomes. This paper presents findings from the Improving Disability Employment Study (IDES): a two-wave survey of 197 DES participants that aims to understand their perspectives on factors that influence access to paid work. Analysis of employment status by type of barrier indicates many respondents experience multiple barriers across vocational (lack of qualifications), non-vocational (inaccessible transport) and structural (limited availability of jobs, insufficient resourcing) domains. The odds of gaining work decreased as the number of barriers across all domains increased with each unit of barrier reported (OR 1.22, 95% CI 1.07, 1.38). Unemployed respondents wanted more support from employment programs to navigate the welfare system and suggest suitable work, whereas employed respondents wanted support to maintain work, indicating the need to better tailor service provision according to the needs of job-seekers. Combined with our findings from the participant perspective, improving understanding of these relationships through in-depth analysis and reporting of DES program data would provide better evidence to support current DES reform and improve models of service delivery.

## 1. Introduction

Evidence suggests that people with disabilities have greater socio-economic and mental health benefits from paid employment than people without disabilities [1,2,3]. Similarly, the negative effects of unemployment (social exclusion, economic disadvantage, poor mental and physical health, housing insecurity) appear to be greater for people with disabilities, potentially due to the existing socio-economic disparities they are often exposed to [2,3,4,5]. Economic arguments highlight that improving employment outcomes for people with disabilities would benefit individuals, families and national-level economic outcomes [6,7]. Yet, the gaps in employment between Australians with and without disability persist [8,9]. Just over half (53%) of working aged Australians with disability are in the labour force, compared to 84% of those without disability. Australians with disability are also more than twice as likely to be unemployed (10% vs. 5%), and experience higher levels of under-employment (11% vs. 8%) [8,9].

Barriers to employment experienced by people with disability are often multifaceted, intertwining individual-level socio-demographic characteristics with vocational, non-vocational and structural barriers to gaining and maintaining work [10,11,12]. Vocational barriers relate to an individual’s level of education and training, skills and qualifications, work experience and work history, and capabilities to undertake job searches and job-related tasks. Longer periods of unemployment can also be considered a vocational barrier with the more time people are unemployed contributing to greater difficulty in gaining work [13,14,15,16]. Non-vocational barriers refer to factors that make it more difficult for an individual to gain or maintain employment or engage with education or skills training. Examples may include health conditions and/or disability, poor mental health, homelessness, experiences of violence and abuse, family responsibilities, lack of access to transport and financial difficulties. Barriers that collectively affect a group disproportionately and contribute to the persistent inequalities often experienced by a group across a person’s lifetime are often referred to as structural, with key examples in relation to people with disability including discrimination and the limited supply of suitable jobs within a labour market [16,17,18]. Insufficient investment and resourcing made available to people with disability to meet their basic and disability-related needs should also be considered as a structural barrier [19,20]. As such, many people with disability have not been provided with equitable opportunities to participate across all life domains on an equal basis with others, undermining their capabilities including in relation to career development and employment [21]. People with disability may also be exposed to compounding structural barriers to employment in relation to their intersectional experiences of gender, ethnicity and indigeneity and geography [22,23].

This paper examines the influence of these barriers on job-seekers with disability engaged with Australia’s Disability Employment Services (DES) program. We present an analysis from the Improving Disability Employment Study (IDES): a two-wave survey that aims to understand DES participant perspectives on factors that influence employment outcomes. We start by providing a contextual overview of the DES program and the current impetus for reforms. This is followed by a summary of the IDES project and an outline of the statistical analysis undertaken for this paper. We then present results on vocational, non-vocational and structural barriers to work disaggregated by employment status, alongside expectations of what supports DES providers should deliver to help participants address barriers to gaining and maintaining work. The discussion positions our findings in the context of current efforts to re-design the DES program and improve employment outcomes for Australians with disability.

### Australia’s Disability Employment Services Program

Australia’s Disability Employment Services (DES) program is the federal government’s specialised welfare program for people whose disability is assessed as their main barrier to employment. For-profit and not-for-profit businesses are contracted by the government to support and monitor people with disability in receipt of income support (and a smaller number of voluntary participants) to ‘actively’ promote their employability and participation in work [24,25,26].

The DES program has undergone considerable reform over the last two decades. The most recent reforms introduced in 2018 were intended to improve employment outcomes by expanding the number of providers within the DES market and incentivising providers to deliver more effective services to participants [27]. Eligibility was expanded to enable more voluntary participants (as opposed to compulsory participants engaged with DES because of their income support mutual obligations), with all DES participants being afforded more choice and control to determine which provider they use and to change providers if they are not satisfied. Educational pathways and outcome payments were enlarged to incentivise providers to support participants to access further education and training to improve their employability [27].

The 2018 reforms were also intended to correct incentives within the funding model that contributed to unintended risk selection behaviour, commonly referred to as ‘creaming and parking’: whereby providers focus attention on easier to place participants in order to maximise financial returns, while providing minimal service to others perceived to be less likely to achieve employment outcomes [28,29]. To address this, the 2018 reforms introduced a new ‘risk-adjusted funding model’, weighting funding on ‘complexity’ of clients and away from servicing and towards educational and sustained employment outcomes achieved. The new funding arrangement was intended to incentivise DES providers to support ‘harder-to-place’ participants to gain and maintain more sustainable employment, ultimately leading to improved employment outcomes across the program [30]. The accompanying DES program grant agreement was initiated 1 July 2018 with an expected end date of 30 July 2023 [31].

Implementing the DES program reform was budgeted at ~AU$850,000 million per annum in 2018 [32]. Yet, the actual cost has escalated to ~AU$1.4 billion per annum in 2021, with revised estimates suggesting that this will increase to ~AU$1.6 billion by 2022 unless program changes are made [33]. Despite this dramatic increase in spending, the reforms have not led to the expected improvement in employment outcomes. Before the 2018 reforms, just over 25% of the ~195,000 DES participants were obtaining employment that lasted at least 26 weeks. Post the reform, less than 25% of the now 315,000 DES participants are obtaining similar employment outcomes. This represents a 12–14% decline in overall outcomes, with a 38% increase in the cost of each 26-week employment outcome achieved ($28,000 pre-reform to $40,000 post-reform) [34,35].

Interestingly, the number of education outcome payments claimed by providers has increased per from $20 million to $148 million per annum, with some providers significantly profiting from enhanced educational pathways and outcomes payments. Concern has been raised as to whether the education and training participants are being enrolled in are actually leading to employment outcomes; particularly, as participants do not need to complete a course or undertake work placement components before providers can claim an education outcome payment [35].

In light of stagnant employment outcomes and burgeoning costs, the government brought forward its planned Mid-term Review of the DES program to 2020, engaging a private consulting firm to assess the program’s current efficacy and evaluate the impact of the 2018 reforms (Australian Government and BCG [35]). Given the national unemployment rate remained relatively stable since the 2018 DES reforms were introduced, the Review did not attribute the stagnation of DES employment outcomes to broader labour market conditions [35]. Rather, the Review attributed the decline to persistent issues with the program, including the limited disability expertise and labour market specialisation of DES providers, which undermines their effective engagement with participants and employers [36,37,38]; the requirement of DES providers to monitor participant mutual obligation compliance making it difficult for staff to develop supportive and positive working relationships with participants [39]; and the complexity of the system and reporting requirements continuing to reduce opportunities for more individualised and innovative service delivery [28,39,40,41,42].

Recommendations emerging from the Review include reducing participant numbers by tightening eligibility based on factors such as age, work capacity, and the relative chance a participant has of obtaining a successful employment outcome within the program [35]. Despite the government collecting extensive program data, the limited analysis reported within the Mid-Term Review makes it difficult for stakeholders to consider how individual-level characteristics and other factors may be influencing DES performance, and whether recommended program reform can be expected to contribute to much needed improved employment outcomes for participants. These gaps in understanding undermine the ability of the government and stakeholders to inform debate on the current re-design of the DES program and shaping of more effective DES models of service delivery. Ultimately, this undermines efforts to improve employment outcomes for Australians with disability. This paper explores factors influencing access to employment from the perspectives of DES participants as well as their expectations of employment programs, as shared through the IDES survey.

## 2. Materials and Methods

### 2.1. The Improving Disability Employment Study

The Improving Disability Employment Study (IDES) aims to improve understanding of factors that promote sustainable and meaningful employment outcomes for people with disabilities. Importantly, it aims to understand these factors from the perspectives of job seekers with disabilities. The IDES project involves the implementation of a two-wave quantitative survey with DES participants 12 months apart. The IDES was conducted in partnership with disability and employment services peak bodies and nine DES providers across Australia [43]. Ethics approval was granted by the University of Melbourne Human Research Ethics Committee (HREC: 1545810).

### 2.2. Survey Design

The IDES survey includes questions across several domains thought to influence and be influenced by employment outcomes including socio-economic characteristics, disability and functioning, current and previous employment and engagement with employment services, health and well-being, housing and transport. The Wave 2 survey covered similar domains to the Wave 1 survey, additionally covering any changes in work or experiences of DES [43].

### 2.3. Implementation

Pilot testing with 32 DES participants was conducted in February 2018, with Wave 1 implemented between April and December 2018 with a total of 337 respondents recruited through the project’s DES partners. Most items were retained between the pilot study and Wave 1 with data combined for the purpose of these analyses. Respondents took approximately 30–45 min to complete the survey. Pilot and Wave 1 respondents were invited to complete Wave 2 of the survey approximately 12 months after completing Wave 1. Data collection for Wave 2 occurred between March 2019 and February 2020 with a total of 197 respondents completing both Wave 1 and Wave 2 of the survey (a 53.4% retention rate). Across both Waves, respondents completed the survey via Computer-Assisted Telephone Interview (CATI) or via an online survey method. Informed consent was collected from all respondents prior to their participation.

### 2.4. Key Exposure Variables

Our key exposure variables were barriers to work, measured at both Wave 1 and Wave 2. Participants were asked if they encountered any of several barriers to work, such as lack of confidence, family responsibilities, and lack of available jobs, with an extended set of barriers included in Wave 2. Participants selected whether the barrier does not affect the work they can do, somewhat affects, or greatly affects. We categorised the barriers into vocational, non-vocational, and structural barrier groupings, reflective of the literature and how these barriers are assessed within Australian employment programs [44]. See Appendix A Table A1 and Table A2 for a full description of the questions and response options across these barriers.

To assess the impact of an increase in the number and intensity of barriers to work, we created four continuous measures of barriers at each wave, reflecting vocational, non-vocational, structural, and all combined barriers. We assigned a score to each of the response options for the barriers to work: does not affect work (score 0), somewhat affects (score 1), and greatly affects (score 2). We summed the scores across barriers for each barrier grouping, noting that respondents did not have to answer every item to be included in the continuous score, as long as they responded to at least one item per barrier grouping (i.e., vocational, non-vocational, structural).

### 2.5. Key Outcome Variables

The key outcome was employment status at Wave 2. Respondents were considered employed if they were currently employed and had therefore either (a) maintained a job held at Wave 1 or (b) gained and maintained a job between Wave 1 to Wave 2. Respondents were considered not employed at Wave 2 if they were not currently in work.

#### 2.5.1. Confounders

We used data from Wave 1 to control for potential confounders of the exposure and outcome variables, including age (18–24, 25–34, 35–49 or ≥50 years), gender (male, female), year 12 completion (yes/no), and ever having worked (yes/no). Additionally, we included an item capturing perception of disability as a barrier to work (does not affect the work I can do, somewhat affects, greatly affects), measured at Wave 2. This barrier was not included in the summed barrier variables used as exposures. In models assessing the association between Wave 2 barriers and employment status at Wave 2, we additionally controlled for employment status at Wave 1 (employed, unemployed).

#### 2.5.2. Other Variables of Interest

To further explore the experiences of respondents who were and were not employed at Wave 2, we examined the expectations respondents had of their DES provider in Wave 1. Participants were asked if they would like their DES provider to offer a number of supports, such as offering suggestions about suitable work, helping participants apply for a job, and providing support once in work. Answers were recorded as yes/no.

In Wave 2, we assessed the support participants received from their DES provider. Participants were asked how good their provider was at providing different kinds of support, such as helping prepare for a job interview or supporting the participant in feeling confident. Responses were coded on a five-point likert scale from very good to very poor.

### 2.6. Statistical Analysis

We began by tabulating descriptive characteristics of Wave 2 respondents’ barriers by employment status. To assess the association between the continuous measures of vocational, non-vocational, structural, and all combined barriers at Wave 1 and employment status at Wave 2, we fit four separate unadjusted and adjusted logistic regression models. In adjusted analyses we controlled for the confounders described above. We repeated this process to investigate the association between the continuous barrier measures at Wave 2 and employment status at Wave 2. Additionally, we examined Wave 1 respondent expectations of their DES provider, and perception of the quality of supports received at Wave 2, by Wave 2 employment status. Finally, to assess the potential for selection bias due to loss to follow-up between wave 1 and wave 2, we used univariate logistic regression to describe the associations between Wave 1 confounders and barriers to work and loss to follow-up at Wave 2. All analyses were performed in Stata v.16 [45].

## 3. Results

This section may be divided by subheadings. It should provide a concise and precise description of the experimental results, their interpretation, as well as the experimental conclusions that can be drawn.

### 3.1. Demographics

Of the 369 IDES Wave 1 participants, 197 (53.4%) responded in Wave 2. Associations between Wave 1 characteristics and loss to follow-up at Wave 2 are shown in Appendix A Table A3. Results do not suggest that age, gender, year 12 completion, or Wave 1 barriers are associated with attrition.

Table 1 describes Wave 1 IDES participant characteristics by Wave 2 employment status. Of the 197 IDES Wave 2 respondents, 39.1% (*n* = 77) were currently employed. Only one-third (35.5%) of female participants were employed at Wave 2, while less than a quarter (24.7%) of respondents aged 50 years and older were employed. One-third (31.2%) of participants who did not complete year 12 and 40.9% of Australian-born participants were employed at Wave 2. Less than one-third—(27.5%) of respondents with no post-school qualifications were employed at Wave 2, while more than half (56.4%) of those with a University degree were employed.

Among individuals with physical disabilities, only 28.8% were in employment at Wave 2, while nearly half (45.7%) of participants with psychological disabilities were. The proportion of individuals who were compulsory DES participants in employment (39.7%) was similar to individuals who were voluntary DES participants (36.4%). A greater proportion of individuals who were with their DES provider for less than six months at Wave 1 were in employment at Wave 2 (50.9%) compared to participants who had been with their provider for 6–12 months (43.8%) or greater than twelve months (31.3%). IDES participants who were employed or who had a job between study waves were asked to report on factors that helped them get their most recent job. A greater proportion of respondents who were employed at Wave 2 reported that they had applied for a job after seeing an advertisement (75.0% versus 25.0% of those not employed); through family or friends (73.3% versus 26.7%); and with the assistance of their employment service (60.9% versus 39.1%).

### 3.2. Barriers to Employment

Across the two waves, a larger proportion of individuals who were not employed reported multiple barriers that greatly affected their ability to find and maintain work. Across the two employment status groups—albeit reported by a greater proportion of those not employed—the most frequent reported vocational barrier to work was not having qualifications or skills, followed by lack of confidence. Poor employment program support was also reported by around one fifth of Wave 2 respondents. While only asked at Wave 2, the most frequently reported non-vocational barrier was health condition/disability, with a far greater proportion of the not employed group reporting this greatly impacted on access to work (62.6% versus 32.5%). This was followed by lack of transport and welfare benefits (noting it was not clear how respondents interpreted items related to welfare benefits, only that it was reported as a barrier). At Wave 1, nearly half (49.6%) of those who were unemployed reported lack of jobs close to where they lived as greatly affecting the work they can do, compared with 31.2% of respondents who were employed. This decreased at Wave 2 to 45.4% and 35.0%, respectively (see Table 2).

Table 3 shows descriptive information for the Wave 1 and Wave 2 continuous barriers measures by Wave 2 employment status. Except for the structural barrier, individuals who were not employed at Wave 2 reported a higher mean and median number of vocational, non-vocational, and combined barriers to work at both Wave 1 and Wave 2.

### 3.3. Regression Analysis of Impact of Barriers

Results of the unadjusted and adjusted logistic regression analyses of the association between Wave 1 barriers to work and Wave 2 employment status are shown in Table 4. The odds ratios (OR) attenuated slightly in adjusted analyses, and we focus on the adjusted results. The OR of 1.40 (95% CI 1.05, 1.86) shows that the odds of being unemployed in DES at Wave 2 are 1.40 times higher (40% higher) for each unit increase in vocational barriers experienced at Wave 1. A one unit increase in non-vocational barriers was likewise associated with increased odds of not being in employment at Wave 2 (OR 1.28, 95% CI 1.06, 1.53). The OR for a one unit increase in a structural barrier may also suggest that increasing structural barriers are associated with greater odds of DES participants not gaining work at Wave 2 (OR 1.52, 95% CI 0.94, 2.44). Considering all Wave 1 barriers together, the odds of being unemployed at Wave 2 increased with each unit increase in barrier reported (OR 1.22, 95% CI 1.07, 1.38).

Results of the unadjusted and adjusted logistic regression analyses of the association between Wave 2 barriers to work and Wave 2 employment status are shown in Table 5. Focusing on the adjusted results, each unit increase in Wave 2 vocational barriers was associated with a 31% increased odds of remaining unemployed (95% CI 1.00, 1.71). Results from the logistic regression models assessing the relationships between a one unit increase in non-vocational barriers (OR 1.02, 95% CI 0.92, 1.13), a structural barrier (OR 1.43, 95% CI 0.85, 2.40), and all barriers considered together (OR 1.04, 95% CI 0.96, 1.13) at Wave 2 on employment outcomes are less conclusive, with confidence intervals that cross the null value. See Table 5.

### 3.4. What DES Participants Want from DES

Table 6 highlights Wave 1 respondents’ expectations of what DES services should do to help them overcome barriers to find and maintain work, compared with Wave 2 perspectives on how DES providers actually performed in delivering on these expectations. Compared to the group of respondents who were employed at Wave 2, a slightly higher proportion of those unemployed wanted help from their DES provider to suggest suitable work (63.3% versus 59.7%), find a training course (49.2% versus 45.5%), and assistance with Centrelink (53.3% versus 42.9%). A greater proportion of those employed wanted help to prepare for interviews (54.6% versus 47.5%), feel confident in their abilities (61% versus 56.7%), and provide support once they had found work (66.2% versus 56.7%) (i.e., this may be to maintain work, increase hours, or find different work). Overall, those employed at Wave 2 were more likely than those not employed to report DES services they had received as good or very good.

## 4. Discussion

Our study demonstrates that many DES participants experience compounding vocational, non-vocational, and structural barriers to gaining and maintaining paid work. Of those that were employed at Wave 2, a greater proportion were younger, had attained higher levels of education, and had spent less time in DES. Employed participants also experienced fewer barriers at both waves. Conversely, for those who were unemployed at Wave 2, a greater proportion were older, less educated and had spent longer periods of time engaged with DES. They were also more likely to report a greater number of barriers to gaining and maintaining work. There appeared to be little difference in employment outcomes between IDES respondents who were compulsory or voluntary DES participants.

A greater proportion of our sample were successful in gaining or maintaining employment when compared to the general DES population (39% compared to ~24%). It could be that in comparison to the general DES population our sample were more job-ready and experienced fewer barriers to employment. More broadly, we cannot rule out that DES participants that do gain employment may do so of their own accord and with limited support from their provider. Indeed, employed respondents were more likely to report they obtained their most recent job through family or friends and after responding to an advertisement, than with the assistance of their employment service. This may indicate a greater level of independence in job-seeking: a similar finding to that reported in parallel IDES qualitative research in which DES participants who gain paid work often reflect that work was found independently of DES [46].

However, we remain unconvinced this means job-seekers with disabilities assessed as having a greater capacity to gain and maintain work should be diverted away from the DES program towards the government’s mainstream employment program, as recommended by the DES Mid-term Review [35]. This proposed program change requires more analysis of how this may influence employment outcomes, with research demonstrating job-seekers with disabilities feel less well-supported within the mainstream employment program [24,46]. More broadly, the even more stringent mutual obligations placed on mainstream employment participants have been found to undermine the well-being and confidence of participants to actively engage in the program and labour market [29,39]. While this proposed policy change may lead to desired cost-savings for the DES program, it is likely that these savings will be shifted to the mainstream employment program and potentially onto to other systems such as health because of the unintended consequences of participants having to work with providers less skilled in working with people with disability.

The DES Review also recommends cost-savings could be achieved through restricting the number of voluntary participants entering DES to increase focus on income support recipients who are mutually obligated to participate in DES. However, we found little difference in employment outcomes between respondents who were compulsorily or voluntarily engaged with DES. Our findings align with studies critical of ‘Welfare to work’ activation policies, particularly in contexts whereby underlying structural barriers to employment such as discrimination and the limited supply of suitable jobs remain unaddressed [39,47,48,49,50,51]. We also question what will happen to excluded voluntary DES participants who are willingly trying to engage with the labour market but need support to do so, particularly given employment plays a vital role in supporting socio-economic and health and well-being outcomes for people with disabilities [1,4,52].

Prior, and subsequent to, the 2018 reforms, commentators on the DES program have argued for more individualised services that better address the complex and multifaceted nature of vocational, non-vocational and structural barriers to employment experienced by DES participants [15,16,28,38,40]. Our analysis examined the impact of these three categories of barriers as individual domains, as well when combined together: demonstrating that multiple barriers across these domains can amplify challenges to gaining and maintaining employment. Beginning with vocational barriers, the lack of qualifications, skills and experience, was reported as greatly affecting the ability to find and maintain work by nearly half of the respondents who were unemployed at Wave 2, compared to just over one fifth of employed respondents. Conversely support to gain qualifications and skills was highly valued across the cohort, with. nearly half of all respondents reporting they wanted their DES provider to help them find a training course, and over half reporting their DES provider has provided good or very good support in this regard.

As highlighted in the Review, DES providers quickly responded to the 2018 reform changes improving financial incentives to encourage participants into further education and training. Yet, there is insufficient evidence that this led to improved employment outcomes for participants. This aligns with concerns in the broader literature that supporting job-seekers to access further education and skills training in and of itself, does not always translate into positive employment outcomes for individuals, or to sufficient increases in employment for people with disability more broadly. Reasons for this may include inconsistent quality of education and training, alongside ineffective support to help people transition from study to paid work alongside structural barriers to employment (e.g., employer discrimination and the limited supply of jobs that meet the needs and aspirations of job-seekers with disability) that remain unaddressed [16,53,54]. Future reform will need to strike the right balance between encouraging providers to support DES participants to access appropriate education and training that will enhance their employment pathways, and unintended financial incentives that encourage providers to funnel participants into training that is less likely to translate into paid employment [35].

More than double the proportion of unemployed IDES respondents reported, compared to employed respondents, lack of confidence greatly affected their ability to find and maintain work. While all respondents wanted support to feel confident in their abilities, unemployed respondents were less likely to report their DES provider had adequately addressed this need. Improving confidence and work self-efficacy is critical to helping individuals navigate and succeed in competitive labour markets [55,56,57,58]. Strategies such as motivational interviewing and recovery-oriented practice within DES could support participants to improve their participant mental health and work-related self-efficacy [46,57,59]. Yet, innovative and more individualised approaches to providing such supports within DES appears to remain rare [35,41,42,46].

Overwhelmingly, respondents reported their health condition or disability as the most common and non-vocational barrier to work. While this is not surprising—given disability as the main barrier to employment is the key eligibility criteria to accessing DES—it does underscore the influence of disability on career development and access to employment [55]. It also underscores that DES need to better support participants to identify employment opportunities that meet their aspirations and disability needs, while ensuring employers are aware of their obligations and available resources to implement reasonable accommodations for employees with disabilities in the workplace. Similarly, as indicated in our results and aligning with research in this area, provision of quality on-the-job supports once work is obtained is similarly valued and required by DES participants to help them maintain employment [38,54,57]. Yet, even if these services and supports are more effectively delivered within a re-designed DES program, their impact will be weakened unless governments can simultaneously address structural barriers to work, such as discrimination, the limited supply of jobs that meet the needs and aspirations of job-seekers with disability, and the insufficient resourcing to support people with disabilities to access required services and supports and develop their work capabilities across their lifetime [19,20,21].

Our results demonstrate the cumulative impact of multiple barriers on the ability of DES participants to find and maintain employment: respondents with more barriers were less likely to be in employment at Wave 2. Unemployed respondents were also more likely to report that barriers persisted across the two waves of the IDES survey. These respondents were more likely to prioritise their DES provider suggesting suitable work, finding a training course, and assisting them to engage with Centrelink. Conversely, employed respondents were less likely to report persistent barriers. They were also more likely to prioritise and report satisfaction with DES services and supports such as help to prepare for interviews, engage with employers about wages and conditions, and on-the-job supports. This may indicate that these participants were not only more ‘job-ready’ but received correspondingly better services and supports from their DES provider. Again, this highlights that more needs to be done to understand how DES funding models may influence provider behaviour and risk selection (i.e., ‘creaming and parking’ less easy to place job seekers) [28,29]. Indeed, the recommended tightening of eligibility for the DES program that may exclude volunteer participants that have somehow been deemed less able to ‘succeed’ in DES could be seen as a systems-level form of ‘creaming’. The need remains for future reform to better align risk-adjusted funding models that more effectively incentivise and resource DES providers to develop individualised models of service delivery that can better tailor supports to meet the needs, priorities and job-readiness of participants.

The Australian Government could greatly improve understanding of the relationships between barriers and employment outcomes through more in-depth analysis and reporting of their own DES program data. Further qualitative research to understand how participants with multiple barriers find and maintain employment, and to explore in-depth service provider attitudes and practices, could help identify elements of good practice to incorporate into future models of service. Such analysis and research would complement our findings and provide more evidence to support current DES program reform, as well as improve models of service delivery. Indeed, such evidence could more broadly inform strategies within and external to DES that help address complex and persistent vocational, non-vocational and structural barriers to work that are all too often experienced by job-seekers with disabilities, so we can finally close the employment gap between Australians with and without disability.

### Strengths and Limitations

To our knowledge, this is the first longitudinal quantitative survey of job seekers with disabilities engaged with Australia’s DES program. It was designed to learn from the perspectives of job seekers themselves, to help strengthen service delivery and answer questions specific to the Australian policy environment. It explored the influence of vocational, non-vocational and structural barriers on gaining and maintaining work: providing a more holistic view of these interactions as experienced by DES participants.

Despite these strengths, given the small size of our sample and modest retention rate it is possible that our findings are not generalizable to the broader DES population. Seven of our DES partners sent the IDES survey as a link through their databases of DES participants: noting that it is difficult to keep such databases up to date given the movement of participants in and out of the system and between different DES providers. While we know approximately 6700 emails were sent, we do not know how many emails were actually received or open by the intended participants. We do know that this approach, however, only elicited 301 responses. This further highlights some of the challenge of externally collecting generalizable data of the DES population, and emphasizes to the value of further examining the relationships between individual-level characteristics, experiences of barriers, and employment outcomes through deeper analysis of the government collected data of DES participants. A larger sample size and a third wave of data would permit more in-depth analysis of how barriers are associated with transitions into and out of employment, and could assist in identifying particular groups of individuals who may benefit from additional, tailored supports from their DES to address the barriers to employment they experience.

There is also potential selection bias among our cohort, with respondents possibly more engaged with DES and the labour market and therefore more motivated to participate in the survey than non-respondents. This may explain why our sample had higher employment outcomes when compared to the broader DES population [34]. It is also likely that other individual-level factors not included in our modelling—such as mental health and well-being- influence employment outcomes [46]. Our survey was also only available in English. This may have limited the inclusion of DES participants from culturally and linguistically diverse communities known to experience intersectional disadvantage in the labour market, and reduced the visibility of these non-vocational and structural barriers in our analysis and results.

Our results indicate that vocational barriers appear to play a more significant role in influencing employment outcomes when compared to non-vocational and structural barriers. This may in part be related to the fact that vocational barriers are more amenable to measurement in this context. Respondents may also be more readily able to reflect on their experiences of vocational barriers as opposed to structural barriers within the labour market (i.e., availability of jobs, being able to utilize knowledge of labour market growth areas). We also note that measures of discrimination were not included in our analysis: were we able to do so, we may have seen a greater influence of structural barriers on outcomes. Further, we acknowledge that insufficient resourcing to meet service and support needs and enable participation across all domains in life can be conceptualized as a structural barrier to employment: especially when this occurs across a person’s lifetime to undermine career development and capabilities to gain and maintain employment [19,20,21]. Further research is required to better understand this issue and experiences within the DES population.

The IDES project was implemented during the 2018 DES reforms. This was a very challenging time for providers, making it more difficult for them to support recruitment and likely reduced the number of respondents participating in our survey. Similarly, the DES population is quite fluid: moving in and out of the program and between different providers. This may have contributed to the fewer than anticipated respondents completing Wave 2 of the survey. Further, because of the small size of the sample we are unable to explore how barriers and employment outcomes differed for example by disability type and age. Nonetheless, our results do enable reflection of factors influencing DES employment outcomes, indicating that a deeper analysis of the broader DES dataset (not publicly available)—alongside future complimentary research—would better inform stakeholder discussion on future DES reform.

## 5. Conclusions

Within our IDES cohort, DES participants with fewer barriers were more likely to gain and maintain paid work. It may be that these participants were more job-ready and gained employment with limited support from their DES provider. Or, they may have received preferential treatment from their DES provider as they were perceived to be more likely to succeed. Conversely, respondents with more barriers were less likely to gain or maintain employment. Whether this is influenced by the interaction of multiple barriers and/or the support they received in DES requires deeper analysis. Strengthening informed debate on future DES reform needs to examine these relationships across the broader DES population. This requires greater and more transparent analysis of the DES dataset held by government, combined with analysis of how similar participants may fair in the mainstream employment program. This is alongside listening to the perspectives of DES participants on the employment program supports they require and value to gain and maintain employment. Ultimately, however, ongoing DES program reform seems somewhat futile unless broader social policies and programs can help prevent and address the complex array of vocational, non-vocational and structural barriers more commonly experienced by job-seekers with disabilities.

## Figures and Tables

**Table 1 ijerph-18-11485-t001:** Wave 1 characteristics of IDES participants by Wave 2 employment status (*n* = 197).

	Employed *n* = 77 *n* (%)	Not Employed *n* = 120 *n* (%)	Total *n* = 197 * *n* (%)
**Gender**			
Male	37 (42.0)	51 (58.0)	88 (100.0)
Female	38 (35.5)	69 (64.5)	107 (100.0)
Other	2 (100.0)	0 (0.0)	2 (100.0)
**Indigenous status**			
Aboriginal and/or Torres Strait Islander	4 (100.0)	0 (0.0)	4 (100.0)
Not Indigenous	73 (38.2)	118 (61.8)	191 (100.0)
**Age category (years)**			
18–24	11 (44.0)	14 (56.0)	25 (100.0)
25–34	19 (42.2)	26 (57.8)	45 (100.0)
35–49	29 (53.7)	25 (46.3)	54 (100.0)
≥50	18 (24.7)	55 (75.3)	73 (100.0)
**Year 12 completion**			
Completed year 12	48 (46.6)	55 (53.4)	103 (100.0)
Did not complete year 12	29 (31.2)	64 (68.8)	93 (100.0)
**Post-school qualifications**			
No additional qualifications	11 (27.5)	29 (72.5)	40 (100)
Certificate I–IV	32 (36.4)	56 (63.6)	88 (100)
Associate degree or diploma	12 (42.9)	16 (57.1)	28 (100)
University degree	22 (56.4)	17 (43.6)	39 (100)
**Country of birth**			
Australia	70 (40.9)	101 (59.1)	171 (100.0)
Elsewhere	7 (26.9)	19 (73.1)	26 (100.0)
**English-speaking background**			
English-speaking	76 (40.4)	112 (59.6)	188 (100.0)
Non-English speaking	1 (11.1)	8 (88.9)	9 (100.0)
**Disability type**			
Physical	19 (28.8)	47 (71.2)	66 (100.0)
Sensory	2 (28.6)	5 (71.4)	7 (100.0)
Psychological	42 (45.7)	50 (54.4)	92 (100.0)
Cognitive	10 (50.0)	10 (50.0)	20 (100.0)
Other or multiple	4 (33.3)	8 (66.7)	12 (100.0)
**DES participation**			
Compulsory	60 (39.7)	91 (60.3)	151 (100.0)
Voluntary	16 (36.4)	28 (63.6)	44 (100.0)
**Length of time with provider**			
<6 months	28 (50.9)	27 (49.1)	55 (100.0)
≥6 months and <12 months	14 (43.8)	18 (56.3)	32 (100.0)
≥12 months	26 (31.3)	57 (68.7)	83 (100.0)
**(Wave 2) main thing that helped you get your most recent job**			
Applied after seeing advertisement	21 (75.0)	7 (25.0)	28 (100)
Through connections from family or friends	11 (73.3)	4 (26.7)	15 (100)
Assisted by employment service	14 (60.9)	9 (39.1)	23 (100)
Recommended by previous employer or work colleagues	3(100)	0 (0.0)	3 (100)
Directly approached an employer	14 (100)	0 (0.0)	14 (100)
Employer approached participant	4 (57.1)	3 (42.9)	7 (100)

* Note that sample sizes vary slightly depending on available participant responses to each item. E.g., only respondents who were currently employed at Wave 2 or had a job since Wave 1 were asked about the main thing that helped them get their most recent job.

**Table 2 ijerph-18-11485-t002:** Wave 1 and 2 barriers greatly affecting ability to find and maintain work, by Wave 2 employment status.

Barriers	Employed (*n* = 77)		Not Employed (*n* = 120)	
	Wave 1 *n* (%)	Wave 2 *n* (%)	Wave 1 *n* (%)	Wave 2 *n* (%)
**Vocational Barriers**				
Not having qualifications, experience, skills	20 (26.0)	16 (21.6)	51 (42.9)	55 (46.6)
Lack of confidence	14 (18.2)	14 (18.2)	46 (38.3)	55 (46.2)
Poor quality employment support program	-	17 (22.7)	-	24 (20.7)
**Non-vocational barriers**				
Health condition/disability	-	25 (32.5)	-	74 (62.2)
Lack of transport	15 (19.5)	12 (15.6)	35 (29.7)	31 (25.8)
Welfare benefits	15 (19.7)	19 (25.3)	29 (25.0)	35 (30.4)
Family responsibilities	6 (7.8)	8 (10.4)	18 (15.1)	19 (16.1)
Caring for others	1 (1.3)	7 (9.2)	13 (10.8)	15 (12.5)
Financial difficulty/debt	-	15 (19.5)	-	27 (23.1)
Lack of access to mental health services	-	14 (18.2)	-	25 (21.2)
Lack of access to health services	-	8 (10.4)	-	19 (16.1)
Housing insecurity	-	11 (14.5)	-	18 (15.0)
Lack of family help	7 (9.1)	8 (10.7)	19 (16.0)	14 (11.8)
**Structural barriers**				
Lack of jobs	24 (31.2)	19 (25.0)	59 (49.6)	54 (45.4)

NB: Additional questions were asked at Wave 2. Please see Appendix A, Table A2 and Table A3 for detailed responses.

**Table 3 ijerph-18-11485-t003:** Descriptive information for Wave 1 and Wave 2 continuous barriers measures.

	Employed				Not Employed			
Wave 1	Mean	Median	Std. Dev.	IQR	Mean	Median	Std. Dev.	IQR
Vocational barriers (range 0–4)	1.83	2	1.24	2	2.44	2.5	1.19	1
Non-vocational barriers (range 0–10)	2.23	2	1.71	2	3.13	2.5	2.38	4
Structural barrier (range 0–2)	1.10	1	0.72	1	1.34	1	0.74	1
Combined all barriers (range 0–15)	5.17	5	2.68	4	6.9	7	3.39	5
Wave 2								
Vocational barriers (range 0–6)	2.26	2	1.69	2	3.11	3	1.60	2
Non-vocational barriers (range 0–18)	4.84	4	3.97	7	5.88	5	4.07	5
Structural barrier (range 0–2)	0.93	1	0.75	1.5	1.24	1	0.79	1
Combined all barriers (range 0–26)	7.97	7	5.66	9	10.24	10	5.32	6

NB: These are observed (as opposed to theoretical) ranges on barrier measures.

**Table 4 ijerph-18-11485-t004:** Unadjusted and adjusted logistic regression analysis, Wave 1 barriers association with Wave 2 employment status.

	Unadjusted			Adjusted *		
	OR	95% CI	*p*-Value	OR	95% CI	*p*-Value
Model 1: Vocational barriers (*n* = 193)						
Wave 1 vocational barriers continuous	1.50	1.18, 1.92	0.001	1.40	1.05, 1.86	0.024
Model 2: Non-vocational barriers (*n* = 193)						
Wave 1 non-vocational barriers continuous	1.24	1.07, 1.45	0.004	1.28	1.06, 1.53	0.009
Model 3: Structural barrier (*n* = 192)						
Wave 1 structural barrier continuous	1.52	1.02, 2.27	0.038	1.52	0.94, 2.44	0.087
Model 4: All barriers (summed, continuous) (*n* = 192)						
Wave 1 all barriers	1.20	1.09, 1.33	<0.001	1.22	1.07, 1.38	0.002

* Adjusted models include age, gender, Wave 2 disability barrier, year 12 completion status, and ever having worked at Wave 1.

**Table 5 ijerph-18-11485-t005:** Unadjusted and adjusted logistic regression analysis, Wave 2 barriers association with Wave 2 employment status.

	Unadjusted			Adjusted *		
	OR	95% CI	*p*-Value	OR	95% CI	*p*-Value
Model 1: Vocational Barriers (*n* = 193)						
Wave 2 vocational barriers continuous	1.40	1.16, 1.70	<0.001	1.31	1.00, 1.71	0.048
Model 2: Non-Vocational Barriers (*n* = 193)						
Wave 2 non-vocational barriers continuous	1.08	1.00, 1.16	0.056	1.02	0.92, 1.13	0.750
Model 3: Structural Barrier (*n* = 191)						
Wave 2 structural barrier continuous	1.69	1.15, 2.47	0.007	1.43	0.85, 2.40	0.179
All Barriers (Summed, Continuous) (*n* = 191)						
Wave 2 all barriers	1.09	1.03, 1.15	0.004	1.04	0.96, 1.13	0.292

* Adjusted models include age, gender, Wave 2 disability barrier, year 12 completion status, ever having worked at Wave 1, and Wave 1 employment status.

**Table 6 ijerph-18-11485-t006:** Wave 1 expectations of DES supports and Wave 2 perspectives on DES supports provided, by employment status.

Wave 1 Expectations			Perspectives on DES Services Provided at Wave 2			
Wave 2 Employment Status						
Supports	Employed	Not Employed	Employed		Not Employed	
	*n* (%)	*n* (%)	Good/Very Good *n* (%)	Neither Good Nor Poor/Poor/Very Poor *n* (%)	Good/Very Good *n* (%)	Neither Good Nor Poor/Poor/Very Poor *n* (%)
Suggest suitable work	46 (59.7)	76 (63.3)	46 (69.7)	20 (30.3)	57 (57.6)	42 (42.4)
Help find training course	35 (45.5)	59 (49.2)	27 (55.1)	22 (44.9)	39 (54.2)	33 (45.8)
Help apply for jobs	44 (57.1)	70 (58.3)	40 (64.5)	22 (35.5)	51 (60.7)	33 (39.3)
Help prepare for interviews	42 (54.6)	57 (47.5)	42 (67.7)	20 (32.3)	44 (57.1)	33 (42.9)
Provide support once I have a job	51 (66.2)	68 (56.7)	47 (72.3)	18 (27.7)	30 (62.5)	18 (37.5)
Support me to feel confident in my ability	47 (61.0)	68 (56.7)	51 (70.8)	21 (29.2)	64 (59.3)	44 (40.7)
Help me participate in decisions	30 (39.0)	41 (34.2)	46 (73.0)	17 (27.0)	60 (60.6)	39 (39.4)
Assistance with Centrelink *	33 (42.9)	64 (53.3)	40 (67.8)	19 (32.2)	48 (49.5)	49 (50.5)
Helping talk to employers about wages and conditions	-	-	30 (62.5)	18 (37.5)	22 (51.2)	21 (48.8)
Support financial costs of training	-	-	31 (66.0)	16 (34.0)	42 (60.0)	28 (40.0)
Help with financial costs of gaining work	-	-	47 (74.6)	16 (25.4)	45 (61.6)	28 (38.4)

Note that sample sizes vary slightly depending on available participant responses. Additional items were included in Wave 2. * Centrelink is the main welfare interface of the Australian Government’s Department of Social Service, responsible for assessing eligibility of welfare programs, referring participants into employment services programs, and administering social security payments (aka. income support payments, pensions).

## Data Availability

Data cannot be shared publicly because of ethical issues regarding consent and potential breaches of confidentiality. Data requests are available from the University of Melbourne Research Data Management team (contact via idesstudy@unimelb.edu.au) for researchers who meet the criteria for access to confidential data.

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
