# Peer review of "Australia’s Disability Employment Services Program: Participant Perspectives on Factors Influencing Access to Work"

_ijerph, 2021, doi:10.3390/ijerph182111485_

Round 1
Reviewer 1 Report
This is a very important topic, and nice source of data to explore something of a conundrum, how ineffective most programs are at getting people with disabilities employed.
My major comment is on the regression model. Being employed in wave 1 is correlated with unobserved characteristics that make the person more employable -- maybe those characteristics lessen the barriers they face (an endogeneity issue)or improve their ability to deal with potential barriers. So I find lumping the two groups together without taking this into account in some way as problematic.
In essence there are two different transitions. Employed to unemployed, and unemployed to employed. I would rather see an estimate of what barriers are correlated with those transitions. Which barriers are more likely to makes someone lose their job? Which barriers are more likely to keep someone from getting a job? They may be the same, but they may be different. And even if not, the magnitude of the effect may be different.
If there were a third wave of the survey -- and that would be great -- you could even estimate a multi-state Markov model that looks at the transitions back and forth between the two states. That would be the most interesting, but of course with two waves that is beyond the scope of this paper.
So I would recommend estimating two logits for both types of transitions.
Some minor points:
1) The authors forgot to put in the descriptive section fr 3.1. It still has the IJERPH formatting guidance.
2) The authors state on line 132 that there have been restrictions tightening eligibility. Isn't that, in effect, a form of creaming?
3) Why isn't prior work experience a confounding factor in their estimation? If it is available it should be included.
4) On line 247 and after --please stick to reporting either employment or unemployment rates. Going back and forth in the text makes it hard to follow.
5) I'm a little concerned that the sample has such a higher employment rate than the general population of program participants. Any ideas? One thought is that respondents to the survey are simply more proactive in general than non-respondents (or face fewer barriers to participating in something) which would be correlated with a higher chance of obtaining employment. Was there a significant non-response rate?
Author Response
Please see attached file for our responses to reviewer comments

Reviewer 2 Report
Thank you for this very important paper. My only recommendation would be to include some information about the intersection/impact of the NDIS on employment for people with disabilities. The paper identifies structural barriers to employment that are 'disability' related. I think this needs some further unpacking in relation to the broader lived experiences of people with disabilities that for many are made more complex due to the inflexibility of supports and services for day to day life. These when discussed alongside the current challenges the NDIS creates for many people with disabilities might give a clearer context to understanding structural 'barriers' - which might be best discussed as systemic barriers. Another framing that might be useful is an intersectional framing of barriers for the lived experience of disability in Australia which must include some mention of indigenous peoples experiences, geographical factors and experiences of women.
Author Response
Please see the attached document for our responses to reviewer comments
